# Anterior Quadratus Lumborum Block and Quadriceps Strength: A Prospective Cohort Study

**DOI:** 10.3390/jcm12113837

**Published:** 2023-06-03

**Authors:** Yuma Kadoya, Nobuhiro Tanaka, Takanori Suzuka, Takayuki Yamanaka, Masato Iwata, Naoki Ozu, Masahiko Kawaguchi

**Affiliations:** 1Department of Anesthesiology, Nara Medical University, Kashihara 634-8522, Japan; kadoyayuuma@naramed-u.ac.jp (Y.K.);; 2Department of Anesthesiology, Yamatotakada Municipal Hospital, Yamatotakada 635-8501, Japan; 3Institute for Clinical and Translational Science, Nara Medical University Hospital, Kashihara 634-8522, Japan

**Keywords:** muscle weakness, muscle strength dynamometer, complications, nerve block, quadratus lumborum block, robot-assisted surgery

## Abstract

The decrease in quadriceps strength after anterior quadratus lumborum block (AQLB) has not been quantified. This prospective cohort study investigated the incidence of quadriceps weakness after AQLB. We enrolled patients undergoing robot-assisted partial nephrectomy, and AQLB was performed at the L2 level with 30 mL of 0.375% ropivacaine. We evaluated each quadriceps’ maximal voluntary isometric contraction using a handheld dynamometer preoperatively and postoperatively at 1 and 4 days. The incidence of muscle weakness was defined as a 25% reduction in muscle strength compared with the preoperative baseline, and “muscle weakness possibly caused by nerve block” was defined as a 25% reduction compared with the non-block side. We also assessed the numerical rating scale and quality of recovery-15 scores. Thirty participants were analyzed. The incidence of muscle weakness compared with preoperative baseline and the non-block side was 13.3% and 30.0%, respectively. Patients with a numerical rating scale ≥ 4 or quality of recovery-15 score < 122, which was classified as moderate or poor, had decreased muscle strength with relative risks of 1.75 and 2.33, respectively. All patients ambulated within 24 h after surgery. The incidence of quadriceps weakness possibly caused by nerve block was 13.3%; however, all patients could ambulate after 1 day.

## 1. Introduction

Minimally invasive surgery causes less postoperative pain and provides rapid recovery; however, acute pain scores are reported to be comparable with those after open nephrectomy, possibly leading to the development of chronic pain [1]. Therefore, the optimization of multimodal analgesia is an urgent issue.

Although typical procedure-specific analgesia protocols, such as enhanced recovery after surgery (ERAS^®^), and procedure-specific postoperative pain management do not exist for robot-assisted laparoscopic partial nephrectomy (RAPN), epidural analgesia, wound infiltration, and peripheral nerve block may have roles in relieving postoperative pain for RAPN. Epidural analgesia is the gold standard for abdominal procedures including nephrectomy. However, it may involve adverse events such as neurological damage, hypotension, epidural hemorrhage, muscle weakness of the lower extremities, and urinary retention, which may be confused with surgical complications. Thus, epidural analgesia is often avoided in patients undergoing RAPN. There is no evidence regarding the efficacy of wound infiltration in laparoscopic nephrectomy or RAPN.

Various studies have investigated the efficacy of peripheral nerve blocks for laparoscopic nephrectomy. Among these, the anterior quadratus lumborum block (AQLB) is a technique intended to provide an analgesic effect for somatic pain from abdominal and hip surgeries [2,3,4]. AQLB was originally conducted at the L4 level, and the effect extended to Th11–12. However, a recent study showed that the approach at the L2 level could be expected to anesthetize from Th6–7 to L1–2 and is more effective for laparoscopic nephrectomy [5,6].

Although the frequency of AQLB is increasing, some reports have indicated that QLB causes a decrease in quadriceps strength, and other studies have reported postoperative muscle weakness in the lower limbs [2,7,8,9] possibly resulting in delayed early ambulation. However, to the best of our knowledge, no study has quantified weakness of the quadriceps muscle after QLB using a handheld dynamometer, which is considered a reliable and valid instrument.

Therefore, we designed a prospective observational study to quantify the quadriceps strength after AQLB by using a handheld dynamometer and assessed its clinical influence under postoperative conditions.

## 2. Materials and Methods

### 2.1. Study Design and Setting

The protocol of this prospective observational study was approved by the research ethics committee of our institution. Informed consent was obtained from all participants. This study was conducted according to the Strengthening the Reporting of Observational Studies in Epidemiology initiative [10], and it adheres to the tenets of the Helsinki Declaration.

The study participants were enrolled between November 2020 and June 2022 at Nara Medical University, with the final follow-up in July 2022.

### 2.2. Participants

Adults between 20 and 75 years old scheduled for RAPN who provided written informed consent were enrolled. The exclusion criteria were as follows: inability to cooperate, dementia, allergy to local anesthetics, chronic use of opioids, coagulation disorder (prothrombin time-international normalized ratio > 1.25, activated partial thromboplastin time > 35 s, platelet count < 10.0 × 10^9^/L), coagulopathy, preoperative muscle weakness or lower limb pain, and body weight < 40 kg or body mass index > 35 kg/m^2^.

### 2.3. Intraoperative Management

All participants received volatile general anesthesia per institutional routine. Fentanyl and remifentanil were administered intraoperatively at the discretion of the anesthesiologist in charge. Wound infiltration with local anesthetics was not permitted, and acetaminophen was administered at the end of surgery. The patients were extubated after conforming the sufficient reversal of the neuromuscular blockade with sugammadex under the train-of-four repetition monitoring. Postoperatively, all participants received intravenous patient-controlled fentanyl analgesia with 0.5 µg/kg/min fentanyl concentration, 1 mL per hour continuous infusion, 1 mL bolus on demand, and a 10 min lock-out interval.

### 2.4. Block Procedure

All blocks were performed using a 20-gauge, 100 mm needle (UNIEVER disposable nerve blockade needle Huber (echogenic), Unisis Corp., Tokyo, Japan). The block operators were YK, NT, and TS, which were familiar with AQLB. The procedure was performed after placement in “nephrectomy position”, which is a lateral decubitus position over a slight table break at the waist.

After skin disinfection, we placed the probe transversely, transitioning laterally from the costal margin on the midaxillary line to the L2 vertebral body and identified the L2 transverse process and the quadratus lumborum muscle. A 20-gauge needle was advanced in-plane through the quadratus lumborum muscle in a lateral-to-medial direction, and saline (1–3 mL) was injected between the quadratus lumborum muscle and the anterior layer of the thoracolumbar fascia to confirm the correct needle tip position. We tried to avoid piercing the fascia of the psoas major muscle and spreading local anesthetic within the psoas major muscle because the local anesthetic would spread to the lumbar plexus through the psoas major muscle [11]. Subsequently, 30 mL of 0.375% ropivacaine was injected to effectuate the AQLB. We recognized the rapid shrinking of the expanded space as successful AQLB.

### 2.5. Measuring Muscle Strength

We evaluated quadriceps strength as maximal voluntary isometric contractions (MVIC) using a handheld dynamometer (MT-100, Sakai Medical Co., Ltd., Fukuoka, Japan) on the day before surgery and on postoperative days (PODs) 1 and 4. The participants were seated with their hips flexed at approximately 85°, knees flexed at 90°, and hands holding the side of the seat (Figure 1). A strap was placed above the ankle joint and adjusted to the correct length. We investigated the incidence of muscle weakness, defined as a 25% reduction in MVIC on POD 1 compared with the preoperative baseline. We also defined “muscle weakness possibly caused by nerve block” as a 25% MVIC reduction compared with the non-block side. Furthermore, we assessed the postoperative course with the numerical rating scale (NRS range, 0–10, with 0 indicating no pain and 10 quadriceps strength the worst pain imaginable) 2 h after surgery and on PODs 1 and 4, and the Japanese version of the quality of recovery-15 (QoR-15 range, 0–150, with a higher score indicating a better quality of recovery) the day before the surgery and on PODs 1 and 4 [12,13,14].

### 2.6. Sample Size Calculation

Because no previous study investigated the incidence of muscle weakness caused by AQLB, we referred to an earlier study that examined the muscle weakness caused by psoas compartment block [15] because the local anesthetics were administered in the similar compartment. This study described that the incidence rate of “no movement” and “active movement only with gravity eliminated” at 6 h after nerve block was 26% and 25%, respectively. We hypothesized that the “no movement” group (26%) or the “no movement” and “active movement only with gravity eliminated” groups (51%) would affect muscle strength after 24 h.

The incidence of these groups ranged from 26 to 51%, and we set the probability of AQLB causing muscle weakness at 35%. We estimated that the incidence of muscle weakness among the 27 patients could be detected with 90% power and a margin of error of ±20% using the Clopper–Pearson confidence interval. The target sample size was set at 30 cases considering a dropout rate of 10%.

### 2.7. Statistical Methods

The primary goal of this study was to estimate the incidence of postoperative muscle weakness on POD 1. We simultaneously assessed muscle weakness compared with preoperative baseline and the strength of the non-block side. The incidence was evaluated as the percentage of participants with muscle weakness, and two-sided 95% confidence intervals (CIs) were determined.

The secondary goals were to investigate the association between muscle weakness and the NRS or QoR-15. We determined the cut-off values of the NRS score and QoR-15 to be ≥4 and ≥122, respectively. We evaluated the relative risk and two-sided 95% CIs.

## 3. Results

We intended to collect 38 participants’ complete data; however, 7 were excluded because they refused to undergo the postoperative muscle evaluation because of pain, postoperative nausea, or hyperpnea. One was excluded because of early discharge on POD 4. Therefore, we included 38 participants between November 2020 and June 2022, and 30 patients were included in the analysis, as shown in Figure 2. The patient characteristics, surgical data, and outcome parameters are presented in Table 1 and Table 2. No significant differences were observed in the patient characteristics or surgical data. The muscle strength of the block side on POD 1 was significantly lower and the NRS scores 2 h after surgery in the muscle weakness group were significantly higher than those in the no-muscle weakness group.

### 3.1. Muscle Strength

A scatter plot of muscle strength of the block side compared with the preoperative baseline and non-block side on POD 1 is shown in Figure 3. The incidence of muscle weakness on POD 1 was 9 out of 30 (30.0%, 95% CI, 14.7–49.4). The incidence of muscle weakness possibly caused by nerve block was 4 out of 30 (13.3%, 95% CI, 3.76–30.7).

### 3.2. Postoperative Pain and Recovery

The NRS scores at each time point are shown in Figure 4. The median values (mean values) of the NRS scores at rest were 2 (1.9) 2 h after surgery; 1 (1.5) on POD 1; and 0 (0.6) on POD 4. The median values (mean values) of the NRS scores for movement were 4 (4.3) on POD 1 and 2 (2.6) on POD 4. Among 30 patients, 11 showed a score of 0 2 h after surgery.

The associations between muscle weakness and NRS or QoR-15 scores are shown in Table 3. Patients with an NRS of ≤4 tended to have muscle weakness on POD 1 and POD 4 with a relative risk of 1.75 (95% CI, 0.44–6.93) and 2.33 (95% CI, 0.58–9.43), respectively. Cases with a QoR-15 score of <122, which was classified as moderate or poor [13] tended to have muscle weakness on POD 1 and 4 with a relative risk of 2.33 (95% CI, 0.58–9.38) and 3.25 (95% CI, 0.86–12.31), respectively.

## 4. Discussion

The incidence of postoperative muscle weakness was 30% and the incidence of muscle weakness possibly caused by nerve block was 13.3% compared with the non-block side. Meanwhile, we defined muscle weakness as a 25% reduction in MVIC compared with the baseline. However, the muscle strength of some patients was distributed around the cut-off values on POD 1. The incidence rates in this study were not definitive. We believe that the significance of this study is that these results provide accurate data on perioperative muscle strength in patients receiving nerve blocks, which may cause muscle weakness.

Although we expected the incidence of quadriceps weakness to be 35 ± 20%, as mentioned above, the incidence in this study was 13.3%. We assume that this low incidence is related to how we performed AQLB. Previous studies have suggested that two pathways of local anesthetics after AQLB cause quadriceps weakness: (1) a pathway posterior to the arcuate ligaments and into the paravertebral space [11,16]; and (2) a pathway into the lumbar plexus through the psoas major muscle. Several cadaveric studies have reported that local anesthetics after AQLB spread into the paravertebral area with a probability of 63–100% [11,17]. This spread into the paravertebral space causes muscle weakness, considering that the paravertebral block is suggested to cause quadriceps motor weakness [18]. Another pathway is through the psoas muscle to the lumbar plexus. A cadaveric study described that local anesthetic in all 10 AQLB procedures had spread consistently to the lumbar plexus and within the psoas major muscle [19]. Another cadaveric study indicated that the lumbar plexus was unaffected if the psoas major muscle was not pierced [11]. Therefore, our approach for avoiding piercing the psoas major muscle in this study may have resulted in the low incidence of quadriceps weakness.

Various factors affect postoperative muscle weakness: pain [20], muscle atrophy [21], inflammation [22,23,24,25], surgical complications or nerve block [26], opioid therapy, and residual neuromuscular block [27]. In this study, patients with NRS scores of ≥4 and moderate or poor QoR-15 scores tended to have muscle weakness. Considering these results, postoperative muscle strength in this study was possibly affected by pain and the quality of recovery. We also compared the muscle strength of the block side with that of the non-block side, excluding factors other than nerve block, such as pain, quality of recovery, or inflammation. Although we compared the postoperative NRS and QoR-15 scores between both groups and there were no significant differences, patients in the muscle weakness group showed a tendency to have more postoperative pain and a lower quality of recovery.

We defined muscle weakness as a 25% reduction in MVIC compared with the preoperative baseline. A previous study showed that handgrip strength was reduced by 16.4% on POD 1 owing to postoperative muscle atrophy [21], and it was considered that quadriceps strength on POD 1 was similarly reduced. Therefore, we set the cut-off value considering other factors, such as postoperative pain and inflammation. We defined a 25% reduction compared with the non-block side as muscle weakness possibly caused by the nerve block because a difference of 10% between sides is physiologically normal in healthy volunteers [28].

The incidence of muscle weakness in this study (13.3%) was slightly lower than that reported in a recent study (16.7%) [8] which was published during the study registration period of the present work. We evaluated muscle strength objectively using a handheld dynamometer; however, the recent previous study documented quadriceps weakness as muscle strength grade 2 out of 5 or less in hip flexion and knee extension 2 h after surgery [8]. This difference in measurements and time points may have affected the results. Although the ERAS® Society recommends early ambulation, and some hospitals encourage patients to walk on the same day as surgery [29], our institutional protocol for RAPN demands that the first ambulation occurs on POD 1 to ensure patient safety. Therefore, we measured muscle strength on POD 1 when the patients ambulated for the first time after surgery.

There are four approaches for QLB based on the injection site: lateral, posterior, anterior, and intramuscular QLB. These approaches were reported to be effective for various surgeries [2,3,4,30]. There has been little evidence of these QLB approaches for laparoscopic nephrectomy; however, AQLB at the L2 level has been reported to be effective for laparoscopic nephrectomy. The original AQLB is performed at the L4 level [31]; however, a previous study showed that its cutaneous sensory blockade is only from T11 to L1 [5]. Therefore, we performed AQLB at the L2 level. In a previous study, the mean NRS scores after RAPN were 5.9, 3.5, and 1.5 on POD 0, 1, and 4, respectively. In this study, the mean NRS scores at rest were 1.9 on POD 0 (11 out of 30 patients showed 0), 1.5 on POD 1, and 0.6 on POD 4, and the NRS scores upon movement were 4.3 on POD 1 and 2.6 on POD 4. These results reflect the effectiveness of AQLB at the L2 level.

We administered 30 mL of 0.375% ropivacaine to all participants, which may have led to a higher incidence of muscle weakness in female participants. To our knowledge, there are no studies that have investigated the minimum effective volume and concentration for AQLB. In previous studies described in a meta-analysis [32], 20–30 mL of 0.2–0.375% ropivacaine was used for QLB, while in a previous study for ATLB at the L2 approach, 20 mL of 0.375% ropivacaine was administered [5]. However, the recent study showed that a larger volume for AQLB contributed to a larger analgesic area [33], supporting our decision to use 30 mL of 0.375% ropivacaine for better analgesia in our surgical settings.

Seven participants were excluded because they refused to provide muscle strength measurements. Among the seven excluded participants, two claimed that nausea and hyperpnea were too severe to participate in the measurements. Four participants refused follow-up owing to wound pain; however, they described low NRS scores (0, 0, 3, and 6 at rest, and 3, 3, 3, and 6 upon movement on POD 1, respectively). It is rational to consider that the participants were hesitant to measure muscle strength because the procedure required maximum strength, which might have caused additional pain. This is a limitation of this study.

There are also other limitations. This was a prospective observational study; hence, a prospective randomized study is required to assess muscle weakness after AQLB compared with placebo in patients undergoing RAPN. However, we investigated the weakness of the block-side quadriceps muscle compared with the non-block side, which can be regarded to show the effect of the nerve block. Furthermore, the influence of postoperative muscle atrophy on muscle strength cannot be excluded. However, our definition of a 25% reduction in MVIC minimized the influence of muscle atrophy in the results.

In conclusion, the incidence of the quadriceps weakness after AQLB on POD 1 was 30.0%, and 13.3% of the total may be affected by AQLB; however, all patients could ambulate on POD 1. A further randomized controlled trial is needed for a clear characterization of AQLB.

## Figures and Tables

**Figure 1 jcm-12-03837-f001:**
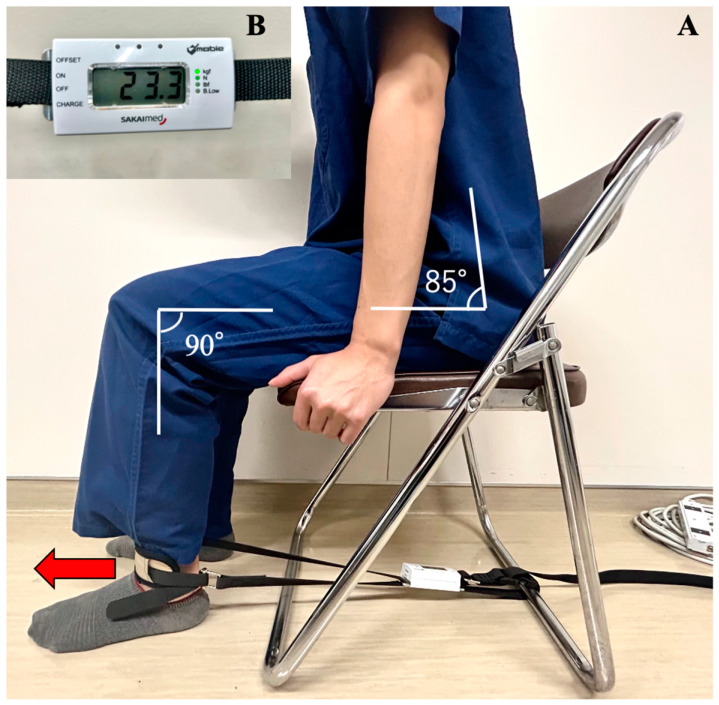
Testing position for strength assessment and the handheld dynamometer. (**A**) One end of the non-elastic belt was fixed on the front of the ankle, and the other was immobilized on the bar of the chair. The red arrow indicates the direction of the force exerted by the participants. The same position and chair were used for all assessments. (**B**) An image of the pull-type handheld dynamometer with non-elastic belts attached to both ends of the device.

**Figure 2 jcm-12-03837-f002:**
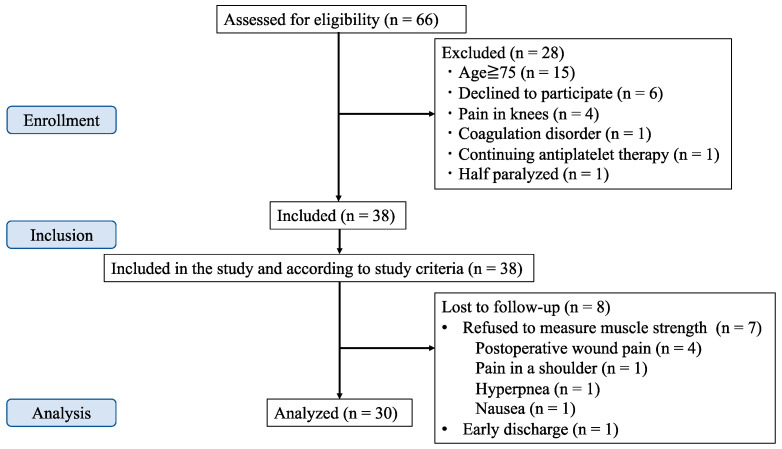
Patient flow diagram.

**Figure 3 jcm-12-03837-f003:**
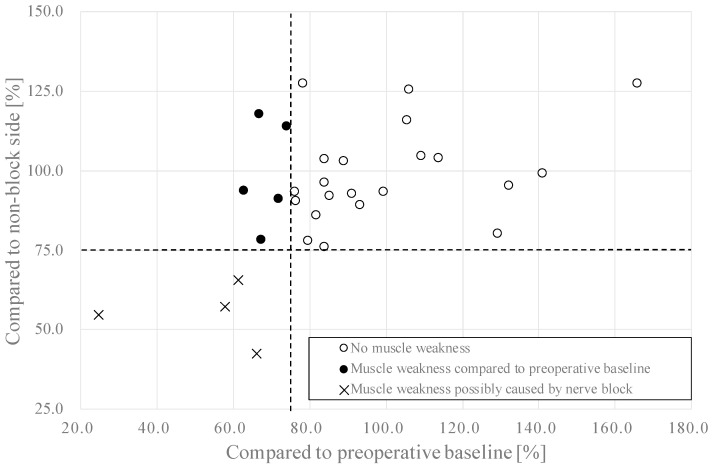
Scatter plot of muscle strength of the block side on postoperative day 1. Dotted lines represent the cut-off values of muscle weakness, which is 25% reduction of muscle strength from the baseline.

**Figure 4 jcm-12-03837-f004:**
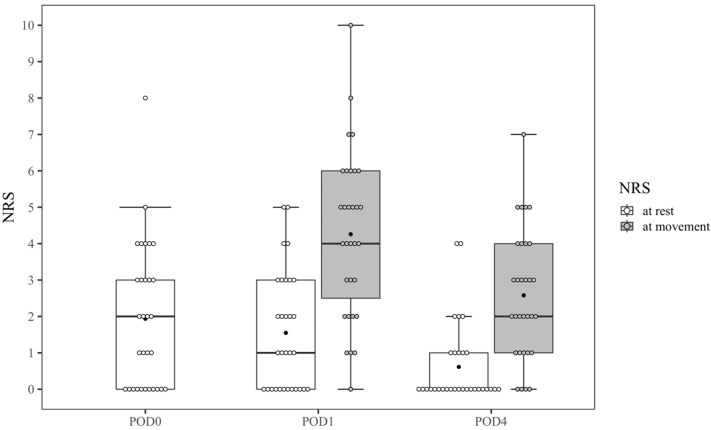
Box plots and dot plots for numerical rating scale scores at each time point. Boxes represent median values (horizontal rule) with the 25th and 75th percentiles (lower and upper limits of boxes, respectively). Error bars indicate the range of non-outlying values. Dots represent the NRS scores of each patient. Black circles represent the mean values of the NRS scores.

**Table 1 jcm-12-03837-t001:** Patient characteristics.

	No-Muscle Weakness Group (n = 21)	Muscle Weakness Group (n = 9)	*p*-Value
Sex			0.10
Male	14 (66.7%)	3 (33%)	
Female	7 (33.3%)	6 (67%)	
Age (years)	64.0 [38, 75]	67.0 [53, 74]	0.70
BMI (kg/m^2^)	24.2 [18.4, 31.8]	22.1 [21.6, 29.1]	0.39
ASA-PS			0.08
1	0 (0%)	2 (22%)	
2	21 (100%)	7 (78%)	
3	0 (0%)	0 (0%)	
4	0 (0%)	0 (0%)	

Values are expressed as medians [min, max] or numbers (proportion). BMI: body mass index, ASA-PS: American Society of Anesthesiologists—physical status.

**Table 2 jcm-12-03837-t002:** Surgical data and outcome parameters.

	No-Muscle Weakness Group (n = 21)	Muscle Weakness Group (n = 9)	*p*-Value
Surgical approach			1.00
Posterior	11 (52.3%)	5 (56%)	
Anterior	10 (47.6%)	4 (44%)	
Ureteral catheter			0.68
With catheter	14 (66.7%)	7 (78%)	
No catheter	7 (33.3%)	2 (22%)	
Duration of surgery (min)	196 [94, 301]	229 [131, 269]	0.39
Muscle strength (kgf)			
Block side			
POD 0	17.8 [6.2, 36.8]	16.7 [10.9–31.4]	0.71
POD 1	18.9 [8.4, 35.2]	12.4 [3.0–21.2]	0.03
POD 4	19.1 [10.0, 42.5]	14.5 [9.1–32.0]	0.20
Non-block side			
POD 0	20.8 [6.0, 35.5]	16.4 [10.6–33.2]	0.59
POD 1	20.3 [6.6, 38.6]	15.6 [5.5–27.1]	0.20
POD 4	15 [9.1, 36.0]	13.1 [10.2–27.0]	0.39
NRS scores at rest			
2 h after surgery	1 [0, 4]	3 [0, 8]	0.02
POD 1	1 [0, 5]	2 [0, 4]	0.80
POD 4	0 [0, 4]	0 [0, 4]	0.42
NRS scores at movement			
POD1	4.5 [0, 10]	5 [3, 6]	0.73
POD4	2 [0, 5]	4 [0, 7]	0.13
QoR-15			
Preoperative	148 [117, 150]	147 [138, 150]	0.20
POD 1	120 [43, 150]	109 [59, 137]	0.22
POD 4	138 [108, 149]	138 [88, 143]	0.44

Values are expressed as medians [min, max] or numbers (proportion). NRS: numerical rating scale, POD: postoperative day, QoR-15: quality of recovery-15.

**Table 3 jcm-12-03837-t003:** Association between NRS or QoR-15 scores and muscle weakness.

		No-Muscle Weakness Group	Muscle Weakness Group	All (n = 30)
POD 1	NRS ≥ 4	13 (65.0%)	7 (35.0%)	20
	NRS < 4	8 (80.0%)	2 (20.0%)	10
	RR	1.75 (0.44–6.93)		
POD 4	NRS ≥ 4	6 (66.7%)	3 (33.3%)	9
	NRS < 4	18 (85.7%)	3 (14.2%)	21
	RR	2.33 (0.58–9.43)		
POD 1	QoR-15 < 122	11 (61.1%)	7 (38.8%)	18
	QoR-15 ≥ 122	10 (83.3%)	2 (16.7%)	12
	RR	2.33 (0.58–9.38)		
POD 4	QoR-15 < 122	2 (50%)	2 (50%)	4
	QoR-15 ≥ 122	22 (84.6%)	4 (66.7%)	26
	RR	3.25 (0.86–12.31)		

RR (risk ratio) is expressed with 95% confidence intervals.

## Data Availability

The data associated with the paper are not publicly available but are available from the corresponding author upon reasonable request.

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
