# Peer review of "Anterior Quadratus Lumborum Block and Quadriceps Strength: A Prospective Cohort Study"

_jcm, 2023, doi:10.3390/jcm12113837_

Round 1

Reviewer 1 Report

Relevant, current-themed, innovative and well-designed work. Some considerations can be made. Several factors that interfere in the evaluation of muscle strength, some referring to the patient, such as age, degree of atrophy, sensitivity to pain, regarding the procedure, such as the technique for performing the blockade (which is operator dependent), and referring to the type of surgery and stress that the procedure offers. Most of these factors have been satisfactorily discussed in the article. As for the description of the anesthetic technique, it could be more detailed, specifying the neuromuscular blocker used and the monitoring of the blockade (did you use TOF?), but anyway, this would not change the results of this research since the first evaluation was carried out on the following day. The patient's position during surgery was not mentioned in the methodology, which could interfere with strength due to some compression or stretching of structures. The choice of the volume of anesthetic used, which in this case was 30ml, was not discussed. Could a volume of 20 ml lead to a lower percentage of quadriceps block? Was it the same anesthesiologist who performed all the blocks? Was there training? In the case of an operator-dependent lock, this information would be interesting. Detailing the anesthetic technique would help in its reproduction in future research involving the subject.

Author Response

The authors would like to thank the reviewer for their constructive critique to improve the manuscript. We have made every effort to address the issues raised and to respond to all comments. Please, find next a detailed, point-by-point response to the reviewer's comments. We hope that our revisions will meet the reviewer’s expectations.

As the reviewer mentioned, muscle strength depends on various factors, such as age, degree of atrophy, sensitivity to pain, the technique for performing the blockade (which is operator dependent), and stress that the procedure offers. We have provided more information concerning anesthesia, especially neuromuscular blockage and the performers of nerve blocks, the intraoperative position, and the methods of nerve block.

#1 As for the description of the anesthetic technique, it could be more detailed, specifying the neuromuscular blocker used and the monitoring of the blockade (did you use TOF?), ….

  • We used rocuronium as neuromuscular blocking agent and mentioned it on P2 L81. In general, we use TOF monitor in all patients in our facilities, and the patients are extubated only when the TOF ratio is > 100%, with sufficient reversal of sugammadex. We also mentioned it on P2 L83–85.

#2 Was it the same anesthesiologist who performed all the blocks? Was there training?

  • We added the information that only designated anesthesiologists performed anterior quadratus lumborum block. We mentioned it on P2 L90–91. These three anesthesiologists are diploma of ‘Japanese Regional Anesthesia Certificate Examination’ and ‘Japanese Board Certificated Anesthesiologist’. These three perform nerve blocks routinely and frequently in our clinical practice.

#3 The patient's position during surgery was not mentioned in the methodology, which could interfere with strength due to some compression or stretching of structures.

  • We added the sentences about the patient position was a lateral decubitus position with a slight table break at the waist. (P2 L91–93)

#4 The choice of the volume of anesthetic used, which in this case was 30ml, was not discussed. Could a volume of 20 ml lead to a lower percentage of quadriceps block?

4) The volume of local anesthetics was determined based on previous studies. However, the effective volume has not been determined yet. Based on our clinical experiences, a higher volume of local anesthetics may affect the incidence of side effects, such as muscle weakness. Nevertheless, this study did not examine the minimum effective volume. We added to ‘Discussion’ the process of determining the amount of local anesthetic. (P10 L 258–265)

Reviewer 2 Report

The article is well written.

There is an issue about sample size calculation. The authors used “an earlier study that examined muscle weakness caused by psoas compartment block [15]”, in which “active movement only with gravity eliminated at 6 hours after nerve block was 26% and 25%, respectively” for sample size calculation. However, the present study is on “quadratus lumborum block”, and the muscle weakness should be much less than psoas compartment block. As mentioned in discussion, a recent study indicated that muscle weakness after quadratus lumborum block was 16.7% [8]. The author should explain why they did not used the result from this study for sample size calculation.

Author Response

The authors would like to thank the reviewer for their constructive critique to improve the manuscript. We have made every effort to address the issues raised and to respond to all comments. Please, find next a detailed, point-by-point response to the reviewer's comments. We hope that our revisions will meet the reviewer’s expectations.

While this point is well taken, a recent study reporting the incidence of muscle weakness after quadratus lumborum block [1] was published during the study-registration period of our work. Therefore, we had no choice but to select the study for psoas compartment block, which is similar to anterior quadratus lumborum block, at that time.

Reference

  1. Shi, R.; Li, H.; Wang, Y. Dermatomal coverage of single-injection ultrasound-guided parasagittal approach to anterior quadratus lumborum block at the lateral supra-arcuate ligament. J Anesth 2021, 35, 307–310, doi:10.1007/s00540-021-02903-1.